# Nanoencapsulation Boosts the Copper-Induced Defense Responses of a Susceptible *Coffea arabica* Cultivar against *Hemileia vastatrix*

**DOI:** 10.3390/antibiotics12020249

**Published:** 2023-01-26

**Authors:** Diego G. Gomes, Karina Sanada, Joana C. Pieretti, Luciana H. Shigueoka, Gustavo H. Sera, Amedea B. Seabra, Halley C. Oliveira

**Affiliations:** 1Department of Agronomy, State University of Londrina UEL, Londrina 86057-970, Brazil; 2Department of Animal and Plant Biology, State University of Londrina UEL, Londrina 86057-970, Brazil; 3Center for Natural and Human Sciences, Federal University of ABC (UFABC), Avenida dos Estados, Santo André 09210-580, Brazil; 4Rural Development Institute of Parana—IAPAR-EMATER (IDR-Parana), Londrina 86047-902, Brazil

**Keywords:** coffee rust, chitosan nanoparticles, biopolymers, disease management, nanotechnology, copper ions

## Abstract

Due to the environmental risks of conventional Cu-based fungicides, Cu-loaded chitosan nanoparticles have been developed as nano-pesticides, aiming to protect plants against different diseases. In this sense, the objective was to verify the effects of chitosan nanoparticles containing Cu^2+^ ions on leaf discs of *Coffea arabica* cv. IPR 100 infected with *Hemileia vastatrix*. The treatments were water as a control (CONT), unloaded chitosan nanoparticles (NP), chitosan nanoparticles containing Cu^2+^ ions (NPCu), and free Cu^2+^ ions (Cu). Different concentrations of NP (0.25; 0.5; 1 g L^−1^) and Cu^2+^ ions (1.25; 2.5; 5 mmol L^−1^) were tested. The severity of the coffee rust was 42% in the CONT treatment, 22% in NP, and 2% in NPCu and Cu. The treatments protected coffee leaves; however, NPCu stood out for initial stress reduction, decreasing Cu phytotoxicity, promoting photosynthetic activity maintenance, and increasing antioxidant responses, conferring significant protection against coffee rust. At low concentrations (1.25 mmol L^−1^), NPCu showed higher bioactivity than Cu. These results suggest that Cu-loaded chitosan nanoparticles can induce a more significant plant defense response to the infection of *Hemileia vastatrix* than conventional Cu, avoiding the toxic effects of high Cu concentrations. Thus, this nanomaterial has great potential to be used as nano-pesticides for disease management.

## 1. Introduction

Coffee rust, caused by the fungus *Hemileia vastatrix* Berk & Broome, is the most important disease in coffee crops, with a worldwide distribution [1,2]. This disease is one of the main limiting factors for producing Arabica coffee, as it causes intense defoliation and the drying of the branches [3,4]. When producers do not adopt control measures, productivity can be reduced by up to 50% depending on environmental conditions and the level of resistance of the genotype [5]. In Brazilian coffee crops, cultivars resistant to the fungus and chemical treatment are the main strategies to control the disease [6,7].

Systemic fungicides have a protective, curative, and eradicating action, sometimes allowing their application with higher rust rates [8]. However, as they have a specific mechanism of action, the combination or alternating the use of systemic fungicides with non-systemic fungicides (such as copper-based) is fundamental to minimizing the selection pressure within the pathogen population, avoiding multiple resistance to systemic fungicides [9]. Cupric fungicides have a multisite action, simultaneously inhibiting more than one process in the fungus cell in different sites or organelles [10]. Multisite fungicides developed in the 1950s and 60s are still recommended for disease management [6]. Taking into account the harmful impacts of copper fungicides on the environment and the risk of resistance to systemic fungicides, new strategies for plant protection have become increasingly necessary due to the limitations and recurrent challenges, in addition to being essential for achieving more sustainable production [11].

In this context, nanotechnology has been studied in agriculture to increase the effectiveness and efficiency of different agrochemicals that are currently available in the market, as well as the creation of new tools, aiming to reconcile greater efficiency (with the consequent decrease in the amount applied) and the reduction in adverse effects on the agrosystem [12]. Developing polymeric nanoparticles as carrier systems for active ingredients has proven to be an effective tool for improving their physicochemical stability. Additionally, they provide a modified release for active ingredients, helping to supply them at the right time and place, minimizing the action of external agents, and reducing losses due to degradation, leaching, and volatilization and, in addition, to unwanted activities on non-target organisms [13].

Among the materials used for the synthesis of nanoparticles, chitosan is a polymer derived from chitin that is considered a very versatile, biocompatible, biodegradable, and non-toxic biopolymer. It has mucoadhesive properties, facilitating the transport of these active compounds across cell membranes, which implies an excellent potential for use in the agrochemical industry [12]. Chitosan nanoparticles have been applied as a seed treatment to release gibberellic acid for growth promotion [14], via the substrate for the controlled release of nitric oxide to protect against abiotic stresses [15,16,17,18,19], and as a seed treatment to release Cu^2+^ ions for nutrition and protection against abiotic stress [20].

In addition to acting as a plant growth-promoting agent, chitosan has antimicrobial properties that are able to induce plant defense responses against pathogens [12]. Another essential characteristic of chitosan is its high affinity for copper compared to other metals, which allows for high encapsulation efficiency [21]. The use of chitosan nanoparticles containing copper ions has been successfully demonstrated in plant species such as tomato [21], maize [22,23,24], and millet [25], either as a plant growth-promoting agent or for its immunomodulatory action and activation of defense mechanisms against biotic and abiotic stresses.

The present study aimed to evaluate the defense responses induced by chitosan nanoparticles containing Cu^2+^ ions on leaf discs of *Coffea arabica* cv. IPR 100 infected with *H. vastatrix*, testing the hypothesis that nano-encapsulated copper provides a better defense when compared to other treatments (Cu^2+^ ions and chitosan nanoparticles without Cu^2+^). To our best knowledge, this is the first report to describe the defense responses of copper-containing chitosan nanoparticles on the leaf discs of coffee infected with *H. vastatrix*.

## 2. Results

### 2.1. Characterization of the Nanoparticles and Encapsulation Efficiency of Cu^2+^ Ions into CS NPs

Chitosan nanoparticles (CS NPs) containing Cu^2+^ ions were successfully synthesized by the ionotropic gelation method, in which the positively charged CS chains were crosslinked with the polyanion TPP [15]. CS NPs containing Cu^2+^ (at a concentration of 5 mmol L^−1^) demonstrated an average hydrodynamic size of 173.0 ± 8.4 nm, a PDI of 0.40 ± 0.02, and a positive zeta potential of 27.4 ± 1.6 mV. The encapsulation efficiency of Cu^2+^ was 57.8%, indicating that 2.9 mmol L^−1^ of Cu^2+^ ions were incorporated into the CS NPs.

### 2.2. Experiment to Assess Phytotoxicity

Regarding the visual aspect of leaf discs, no symptoms of phytotoxicity were observed 40 days after treatments at any of the tested concentrations of the formulations: unloaded chitosan nanoparticles (NP); chitosan nanoparticles containing Cu^2+^ ions (NPCu) and Cu^2+^ ions (Cu). It was possible to verify increases in the chlorophyll *a* content provided by the NP and NPCu treatments, highlighting gains of 33% (NP 0.25 g L^−1^) and 50% (NPCu 2.5 mmol L^−1^) compared to CONT (water as a control) (Figure 1a). On the other hand, none of the tested Cu treatment concentrations influenced the chlorophyll *a* content (Figure 1b). All treatments provided an increase in the chlorophyll *b* content compared to the CONT, highlighting gains of 23% (NP 0.25 g L^−1^), 26% (Cu 5 mmol L^−1^), and 42% (NPCu 2.5 mmol L^−1^) (Figure 1c,d). Moreover, it was possible to verify a reduction in the chlorophyll *a*/chlorophyll *b* ratio induced by increasing concentrations of the formulations, mainly in the Cu treatment (Figure 1e,f).

The NP and NPCu treatments provided significant gains in the total chlorophyll content, highlighting an increase of 28% (NP 0.25 g L^−1^) and 46% (NPCu 2.5 mmol L^−1^) to the CONT (Figure 2a). In contrast, the Cu treatment did not influence the total amount of chlorophylls (Figure 2b). Regarding the carotenoid content, an increase of 13% (NP 0.5 g L^−1^) and 17% (NPCu 2.5 mmol L^−1^) was observed, followed by a slight reduction in the highest concentrations of NP and NPCu compared to CONT (Figure 2c). On the other hand, the Cu treatment promoted a reduction in the carotenoid content in all the tested concentrations, with a 23% reduction caused by the highest concentration (Cu 5 mmol L^−1^) (Figure 2d). All concentrations of NPCu increased the total chlorophylls/carotenoids ratio compared to the CONT, highlighting a gain of 55% in the highest concentration (NPCu 5 mmol L^−1^). Despite the increases of 22% (NP 0.25 g L^−1^) and 52% (Cu 2.5 mmol L^−1^), there was a reduction in this parameter by higher concentrations of NP and Cu (Figure 2e,f).

### 2.3. Evaluation of the Disease Progression

By evaluating the visual aspect of the leaf discs, it was possible to verify a high incidence of the disease in the CONT and NP treatments (0.25 and 0.5 g L^−1^) at 15 days after infection (DAI) (Appendix A). In the other treatments, the incidence of disease on the leaf discs at 15 DAI was considerably lower (Appendix A). In general, sporulation started at 20 DAI, intensifying at 25 DAI in the CONT and NP, as well as in NPCu and Cu treatments at the lowest Cu concentration (1.25 mmol L^−1^). On the other hand, the protective effect conferred by NPCu and NP treatments (2.5 and 5 mmol L^−1^) was visible over the 40 DAI (Appendix A).

In the CONT and NP treatments (0.25 and 0.5 g L^−1^), 97% of the leaf discs showed the first symptoms of infection by *H. vastatrix*. In contrast, NP (1 g L^−1^) and Cu (1.25 mmol L^−1^) showed 44% of the leaf discs as having disease incidence, while in the NPCu treatment (1.25 mmol L^−1^), the incidence was 28%. The treatments of NPCu and Cu (2.5 and 5 mmol L^−1^) showed the lowest incidence of the disease (less than 2%) (Figure 3a).

Sporulation occurred in 94% of the leaf discs of CONT and NP treatments (0.25 and 0.5 g L^−1^). The NP (1 g L^−1^), NPCu (1.25 mmol L^−1^), and Cu (1.25 mmol L^−1^) treatments did not differ, showing sporulation in 25% of the leaf discs. In the NPCu and Cu treatments (2.5 and 5 mmol L^−1^), sporulation was practically null, with less than 1% of leaf discs showing sporulation at 25 DAI (Figure 3b). 

Regarding disease severity at 40 DAI, the observed pattern was the same. On average, the CONT and NP treatments (0.25 and 0.5 g L^−1^) showed 35% of the injured leaf disc area. The NP (1 g L^−1^), NPCu (1.25 mmol L^−1^), and Cu (1.25 mmol L^−1^) treatments did not differ from each other, presenting around 5% of the injured leaf disc area. In NPCu and Cu treatments (2.5 and 5 mmol L^−1^) and the damaged leaf area was less than 1% (Figure 3c).

The NPCu and Cu treatments positively affected the chlorophyll *a* content of the infected leaf discs, with increases of 107% (Cu 2.5 mmol L^−1^) and 133% (NPCu 2.5 mmol L^−1^) compared to the CONT, while NP did not affect this parameter (Figure 4a,b). All formulations provided higher chlorophyll *b* contents than the CONT, highlighting the increases by 95% (NP 1 g L^−1^), 129% (Cu 2.5 mmol L^−1^), and 215% (NPCu 2.5 mmol L^−1^) (Figure 4c,d). In addition, only the NPCu and Cu treatments decreased the chlorophyll *a*/chlorophyll *b* ratio (Figure 4e,f).

All formulations increased the total amount of chlorophyll compared to the CONT, highlighting the increases by 82% (NP 1 g L^−1^), 113% (Cu 2.5 mmol L^−1^), and 170% (NPCu 2.5 mmol L^−1^) (Figure 5a,b). Similarly, all treatments provided higher amounts of carotenoids than the CONT (Figure 5c,d), as well as increasing the total chlorophylls/carotenoids ratio (gains of 55% in NP 1 g L^−1^, 142% in Cu 5 mmol L^−1^, and 155% in NPCu 5 mmol L^−1^, to the CONT) (Figure 5e,f).

### 2.4. Evaluation of Oxidative Stress and Antioxidant Response

Significant changes in the content of hydrogen peroxide (H_2_O_2_) and malondialdehyde (MDA) were induced by the applied treatments (Figure 6). At 5 DAI, NP treatments (0.25 and 0.5 g L^−1^) presented, on average, 13% more H_2_O_2_ compared to the CONT, while Cu treatments (1.25 and 2.5 mmol L^−1^) did not affect this parameter. In contrast, the NPCu treatments (1.25 and 2.5 mmol L^−1^) reduced the H_2_O_2_ content in infected leaf discs by 20%. At 15 DAI, only NP (0.5 g L^−1^) differed from the CONT, maintaining the highest H_2_O_2_ content (Figure 6a). Regarding the MDA content, only the Cu treatment (2.5 mmol L^−1^) did not differ from the CONT, with higher MDA concentrations observed in the other treatments at 5 DAI. At 15 DAI, Cu (1.25 and 2.5 mmol L^−1^) and NP (0.5 g L^−1^) treatments did not show significant differences to the CONT, while the NP treatment (0.25 g L^−1^) induced an increase of 46% in the MDA content. In contrast, the NPCu treatments (1.25 and 2.5 mmol L^−1^) promoted an average reduction of 37% in the MDA content compared to the CONT (Figure 6b).

Regarding the activity of antioxidant enzymes, only the NP treatment (0.25 g L^−1^) induced a 27% reduction in superoxide dismutase (SOD) activity at 5 DAI. At 15 DAI, NP treatments (0.25 and 0.5 g L^−1^) did not show significant differences compared to CONT, while NPCu and Cu treatments (1.25 and 2.5 mmol L^−1^) induced an average increase of 22% in SOD activity (Figure 7a). At 5 DAI, all treatments differed from the CONT, showing reduced catalase (CAT) activity. At 15 DAI, there was an average reduction of 20% in CAT activity in the NPCu (1.25 mmol L^−1^) and Cu (1.25 and 2.5 mmol L^−1^) treatments compared to the CONT (Figure 7b).

The NP (0.25 and 0.5 g L^−1^) and Cu (1.25 mmol L^−1^) treatments induced an increase in ascorbate peroxidase (APX) activity compared to the CONT at 5 DAI. In contrast, NPCu treatment (2.5 mmol L^−1^) promoted a 54% reduction in enzyme activity. At 15 DAI, only the NPCu treatment (1.25 mmol L^−1^) failed to show significant differences compared to CONT, with an increase in the APX activity observed in the other treatments (Figure 7c). For peroxidase (POD) activity, at 5 DAI, it was possible to observe an average increase of 211% in NPCu (1.25 mmol L^−1^) and Cu (2.5 mmol L^−1^) treatments compared to the CONT. At 15 DAI, a rise of 60% in POD activity was induced by the Cu treatment (1.25 mmol L^−1^), 160% by NPCu (1.25 mmol L^−1^), 180% by NP (0.25 g L^−1^), 250% by NPCu (2.5 mmol L^−1^), and 310% by Cu (2.5 mmol L^−1^), compared to the CONT (Figure 7d).

### 2.5. Evaluation of Chlorophyll a Fluorescence and Activity of PAL and PPO Enzymes

Regarding the fluorescence of chlorophyll *a*, all treatments showed lower basal fluorescence (F_0_) values than the CONT at 5 DAI. The reduction in F_0_ was more intense (around 18%) in NPCu (2.5 mmol L^−1^) and Cu (1.25 mmol L^−1^) treatments. At 15 DAI, NP (0.25 g L^−1^) and NPCu (1.25 mmol L^−1^) treatments did not affect the F_0_, while the other treatments showed higher F_0_, especially in the NP (0.5 g L^−1^) and Cu (2.5 mmol L^−1^) treatments, with an average increase of 21% compared to the CONT (Figure 8a).

NP (0.25 g L^−1^) and NPCu (1.25 and 2.5 mmol L^−1^) treatments promoted higher values of the maximum activity of photosystem II (F_v_/F_m_) than CONT at 5 DAI. On the other hand, Cu treatments (1.25 and 2.5 mmol L^−1^) showed a lower F_v_/F_m_. At 15 DAI, an increase in F_v_/F_m_ was induced by NP (0.25 and 0.5 g L^−1^) and NPCu (1.25 mmol L^−1^) treatments, while Cu (2.5 mmol L^−1^) continued to promote a reduction in F_v_/F_m_ compared to the CONT (Figure 8b).

For the relative electron transport rate of photosystem II (rETR), only NPCu treatments (1.25 and 2.5 mmol L^−1^) were significantly increased at 5 DAI compared to the CONT. At 15 DAI, the lowest concentrations of NP and NPCu induced higher rETR values than the CONT. On the other hand, an 18% reduction in rETR was observed in the Cu treatment (2.5 mmol L^−1^) compared to the CONT (Figure 8c). 

At 5 DAI, all formulations at the lowest concentrations (0.25 g L^−1^ or 1.25 mmol L^−1^) and the NP treatment (0.5 g L^−1^) increased the phenylalanine ammonia-lyase (PAL) activity by nearly 100% compared to the CONT. At 15 DAI, NPCu (2.5 mmol L^−1^) and Cu (1.25 mmol L^−1^) treatments maintained PAL activity equal to CONT, while the other formulations reduced the enzyme activity (Figure 9a). Regarding the polyphenol oxidase (PPO) activity at 5 DAI, an average increase of 40% was observed in the NP (0.5 g L^−1^) and NPCu (1.25 mmol L^−1^) treatments compared to the CONT. Furthermore, a 14% increment was induced by the Cu treatment (1.25 mmol L^−1^). At 15 DAI, there were no significant differences among treatments in PPO activity (Figure 9b).

## 3. Discussion

### 3.1. Characterization of the Nanoparticles

CS NPs containing Cu^2+^ were found to be at the nanoscale with moderate polydispersity and a positive zeta potential due to the cationic nature of chitosan. The magnitude of zeta potential suggests the formation of a stable colloidal dispersion. These results are in accordance with our previous publication [20]. The encapsulation efficiency of Cu^2+^ (initial concentration 5 mmol L^−1^) was found to be lower (57.8%) than the encapsulation efficiency of Cu^2+^ at the initial concentration of 1 mmol L^−1^ (97%), as previously reported by Gomes et al. [20]. As expected, these Cu^2+^-containing nanoparticles are spontaneously formed by the electrostatic interactions of Cu^2+^, chitosan chain, and TPP, with an optimum concentration for each component. It should be noted that, for plants treated with free Cu^2+^ ions, we selected a concentration of 5 mmol L^−1^, considering both encapsulated and non-encapsulated ions in NPCu.

### 3.2. Phytotoxicity Assay

Formulations containing CS NPs (NP and NPCu) promoted a dose-dependent effect, highlighting the increased content of chlorophylls and carotenoids at the intermediary concentration (Figure 1 and Figure 2). In contrast, Cu treatments decreased the carotenoid content in all tested concentrations, which indicated phytotoxicity. Choudhary et al. [24] reported that CS NPs containing Cu^2+^ reduced the content of chlorophylls *a* and *b* in maize leaves as the nanoparticle concentration increased. In the present study, the increment in chlorophyll levels by NP and NPCu treatments may be linked to the activation of genes related to the synthesis of this pigment and promoted by chitosan [12]. The alterations in the pigment ratios (decrease in chlorophyll *a*/chlorophyll *b* and an increase in chlorophyll/carotenoid) indicate a maximization of light capture under the low-light environment of the experiment [26]. Regarding the content of the evaluated pigments, NPCu provided the most significant gains, which can be related to the biostimulant effects of chitosan and the mitigation of the copper excess phytotoxicity through the controlled release of the element by the nanoparticles.

### 3.3. Disease Progression

The NP, NPCu, and Cu treatments effectively protected the leaf discs, reducing the incidence, sporulation, and severity of *H. vastatrix*. However, the superiority of NPCu and Cu treatments compared to NP and CONT was easily verified (Appendix A, and Figure 3). Copper in its conventional form has been used for years as an active ingredient in contact fungicides to control pathogenic fungi, these products being composed of copper sulfate, copper hydroxide, copper oxychloride, or copper carbonate [27].

NPCu provided the highest concentrations of chlorophyll in the leaf discs, and both NPCu and Cu provided a similar dual effect on the carotenoid content (Figure 4 and Figure 5). For chlorophylls, there was a more significant protective effect which was conferred by copper nanoencapsulation in all the tested concentrations, which indicates improved bioactivity resulting from the association between chitosan and copper. Saharan et al. [21] demonstrated that CS NPs have porosity, which allows copper adhesion. The porous architecture is responsible for the gradual release of copper from the nanostructure, making the contact of plant cells with copper lasting. This structural feature gives chitosan a higher affinity for copper than other metals. CS NPs containing copper have shown high antifungal activity against *Alternaria alternata*, *Macrophomia phaseolina* and *Rhizoctonia solani* [28], *Alternaria solani*, *Fusarium oxysporum* [21], *Curvularia lunata* [24], and *Pyricularia grisea* [25]. In NP treatments, the protection is associated with the natural characteristics of chitosan. In addition to acting as a plant growth-promoting agent, chitosan has antimicrobial properties and can induce a plant defense response against pathogens. In addition to versatility, chitosan is biocompatible, biodegradable, and can be mucoadhesive [12].

The adverse effects of increasing concentrations of all formulations were observed in the assays, especially in treatments containing copper (NPCu and Cu). Thus, the highest concentrations of NP (1 g L^−1^), NPCu (5 mmol L^−1^), and Cu (5 mmol L^−1^) treatments were removed in subsequent experiments to evaluate oxidative damage, enzymatic activity, and chlorophyll *a* fluorescence.

### 3.4. Oxidative Stress and Enzymatic Activities

The slight increase in the MDA content at 5 DAI indicates the initial induction of oxidative stress by most treatments (Figure 6). However, H_2_O_2_ levels were normal or even reduced compared to CONT in the NPCu and Cu treatments, suggesting that protective mechanisms were induced from 5 DAI or earlier. They might have contributed to decreasing the MDA content in these treatments at 15 DAI.

In contrast, increased H_2_O_2_ levels in the NP treatments indicate a more significant induction of oxidative stress. It can also be associated with the signaling function of H_2_O_2_ triggered by the CS NPs. Indeed, the maintenance of high levels of H_2_O_2_ in the NP (0.5 g L^−1^) treatment at 15 DAI, concomitant with normal levels of MDA, is evidence of the signaling function of H_2_O_2_. According to Hidangmayum et al. [29], in addition to having biostimulant characteristics, chitosan elicits plant defense responses against stress. Its application can induce the synthesis of different intracellular messengers that act in stress signalings, such as H_2_O_2_ and nitric oxide. In addition, chitosan is capable of forming complexes with metals [29]. This ability may have contributed to the induction of more significant oxidative stress in treatments with unloaded CS NPs, causing ionic imbalance and compromising the functionality of other compounds in plant metabolism. In a previous study, Gomes et al. [20] observed that seed priming with CS NPs containing Cu^2+^ ions did not significantly increase H_2_O_2_ and MDA levels in soybean seedlings. However, the treatment with unloaded CS NPs practically doubled the levels of both markers of oxidative stress.

As for the activity of antioxidant enzymes (Figure 7), there were differences in the defense responses induced by each treatment. For SOD, all copper-containing formulations increased enzyme activity at 15 DAI compared to CONT and NP, which could be related to the fact that SOD is a metalloenzyme that has copper in the composition of one of its isoforms [30]. Accordingly, the treatment with Cu^2+^-containing CS NPs has also been shown to induce SOD activity in maize leaves [24] and soybean seedlings [20]. Moreover, by evaluating the effect of a nanocomposite of selenium and copper on *Alternaria solani*-infected tomato plants, Quiterio-Gutiérrez et al. [31] verified an increase in leaf SOD activity which was concomitant with the increase in copper concentration. 

Although the H_2_O_2_ formed by the action of SOD during the neutralization of the superoxide anion can be metabolized by the action of CAT and APX [30], the fine modulation of H_2_O_2_ was performed mainly by APX, allowing the control of this molecule in the μM range and compartments other than peroxisomes [32]. Indeed, the APX activity tended to be induced by most treatments, while CAT activity was maintained or inhibited (Figure 7). Moreover, the low CAT activity in Cu and NPCu treatments can be related to the inhibition of the enzyme by metal ions, as observed by Gomes et al. [20].

Interestingly, there was an increase in POD activity in treatments that showed low CAT and APX activities at 5 DAI, which provides evidence for the induction of different antioxidant enzymes by the applied treatments. At 15 DAI, POD activity was significantly increased upon almost all treatments containing copper, especially at the highest concentration. In coffee plants, the induction of enzymes related to cell wall lignification, such as POD, is one of the primary forms of protection against infection by *H. vastatrix* [33,34,35]. At 5 DAI, the NPCu treatment (1.25 mmol L^−1^) induced higher POD activity than NP (0.25 and 0.5 g L^−1^) and Cu (1.25 mmol L^−1^) treatments. At 15 DAI, NPCu (1.25 and 2.5 mmol L^−1^) also promoted a significant increase in POD activity compared to the CONT, while Cu potentiated the enzyme activity only at the highest concentration (2.5 mmol L^−1^). These results reinforce the fact that NPCu shows greater efficiency in inducing protection against *H. vastatrix* infection compared to the application of chitosan and copper separately, as observed for the protection of soybean seedlings against abiotic stress [20] and the defense of maize leaves against fungal infection [24].

PAL activity was induced during pathogen establishment in the host tissues [33]. Here, all formulations at the lowest concentration promoted an increase in PAL activity at 5 DAI (Figure 9). Similarly, the most significant effects on PPO activity were observed at 5 DAI, highlighting the increased activities in NP (0.5 g L^−1^), NPCu (1.25 mmol L^−1^), and Cu (1.25 mmol L^−1^) treatments. Choudhary et al. [24] evaluated the PAL and PPO activities in maize leaves infected with *Curvularia lunata* at 24 h after treatment with the formulations. The authors verified that NPCu and Cu treatments induced the PPO activity, while only NPCu promoted increased PAL activity [24].

### 3.5. Photosynthetic Activity

Chlorophyll *a* fluorescence is a widely used technique to study the effects of different types of stress on photosynthetic activity [36]. According to Figure 8, at 5 DAI, the lowest values of F_0_ provided by all treatments compared to CONT indicated the reduction in damage to photosystem II. Furthermore, the highest values of F_v_/F_m_ and rETR suggest that NPCu treatments up to 5 DAI improved the photosystem II activity compared to all other treatments. On the other hand, the reduction in F_v_/F_m_ induced by Cu treatments indicates the limitation of photosystem II caused by the action of *H. vastatrix* and/or a high concentration of Cu^2+^.

At 15 DAI, an increase in F_0_ was induced mainly by the Cu treatment (2.5 mmol L^−1^). Furthermore, Cu (2.5 mmol L^−1^) was the only treatment that showed a significant reduction in F_v_/F_m_ and rETR compared to CONT, which corroborates the phytotoxic effect caused by high Cu^2+^ concentrations. The decrease in the F_v_/F_m_ ratio, concomitant with an increase in F_0_, indicates damage to photosystem II, which has been observed in plants under different types of stress, including coffee under high aluminum [37] and drought [38].

### 3.6. CS NPs Containing Cu^2+^ ions as Potential Nanofungicide for Coffee Rust Management

The present study was the first to show the effect of nanoencapsulation on Cu^2+^ ions as a strategy to prolong/intensify the protection conferred by copper against the infection of *H. vastatrix* in a susceptible coffee cultivar (*Coffea arabica* cv. IPR 100). According to the results obtained, it was possible to verify that the induced responses are dose-dependent, with harmful effects caused by the highest concentrations tested. However, NPCu treatments decreased Cu phytotoxicity and showed more positive responses on the leaves by the association between chitosan and copper, especially at low concentrations (1.25 mmol L^−1^). Without the infection of *H. vastatrix*, treatments containing CS NPs (NP and NPCu) stimulated chlorophyll accumulation in leaf disks. In contrast, treatments with free Cu^2+^ ions did not promote significant gains in chlorophyll and reduced carotenoid content. 

Although all treatments effectively decreased the incidence, sporulation, and severity of the disease caused by *H. vastatrix*, the treatments containing copper (NPCu and Cu) promoted a more effective protective effect. However, NPCu (2.5 mmol L^−1^) was the most effective in enhancing the chlorophyll concentrations in infected leaf discs. In general, the treatments promoted the activation of enzymatic antioxidant mechanisms, but NPCu treatments were more effective in mitigating oxidative stress and maintaining photosynthetic activity over the days after infection. Although treatments with free Cu^2+^ ions are efficient in protecting coffee leaves against *H. vastatrix*, they promote adverse effects on photosystem II. In addition, the results showed the more significant bioactivity of the association between chitosan and copper compared to the application of chitosan and copper separately, mainly when evaluating the lowest concentrations of the formulation.

Overall, these results suggest that the CS NPs containing Cu^2+^ are nanomaterials capable of inducing a more significant plant defense response to the infection of *H. vastatrix*, avoiding the toxic effects of high Cu concentrations. Thus, this formulation has great potential to be used as a nano-pesticide for other fungal diseases of different plant species that are controlled by applying copper-based products. Using copper-based nanoparticles to control coffee rust could also reduce the amount of metallic copper released per hectare due to the improved delivery and greater contact area of the nanoparticles with the leaves, providing better coverage. A benefit would be a chemical control strategy for coffee rust in line with the new legislation implemented in several countries, which aims to reduce the impacts caused by metallic compounds in agriculture [7].

## 4. Materials and Methods

### 4.1. Chemicals

Chitosan (75% deacetylation, low molecular weight), copper chloride II (CuCl_2_), and sodium tripolyphosphate (TPP) were purchased from Sigma–Aldrich (St. Louis, MO, USA). Fresh Milli-Q^®^ water was employed in the synthesizing and characterizing of chitosan nanoparticles (CS NPs).

### 4.2. Synthesis of CS NPs Containing Cu^2+^ Ions

As previously reported, CS NPs containing Cu^2+^ ions were synthesized by the ionotropic gelation method [20]. In brief, chitosan (1 mg mL^−1^) and CuCl_2_ were dissolved in an acetic acid solution (1%) under vigorous stirring for 1.5 h. Subsequently, a TPP solution (0.6 mg mL^−1^) was dropwise in the chitosan/Cu^2+^ solution. The process led to the formation of CS NPs containing 5 mmol L^−1^ of Cu^2+^. For unloaded CSNPs, a similar procedure was performed without adding CuCl_2_.

### 4.3. Characterization of the Nanoparticles and Encapsulation Efficiency of Cu^2+^ Ions into CS NPs 

The hydrodynamic size, polydispersity index (PDI), and zeta potential of Cu^2+^-loaded CS NPs were analyzed by dynamic light scattering, employing a Nano ZS Zetasizer (Malvern Instruments Co, Malvern, UK) in folded capillary zeta cells with a 10 mm path length and a fixed angle of 173° [39]. Measurements were performed in triplicate.

The encapsulation efficiency of the Cu^2+^ ions into CS NPs was evaluated, as previously reported by Gomes et al. [20]. Briefly, free Cu^2+^ ions (non-encapsulated) were separated from encapsulated Cu^2+^ ions using Amicon ultra centrifugal filters (molecular weight cut off 10 kDa, Millipore, Darmstadt, Germany). Free Cu^2+^ ions were quantified by inductively coupled plasma mass spectrometry (ICP-MS) (Agilent 7900, Hachioji, Japan). The 50 µL aliquots of free Cu^2+^ ions were acidified with HNO_3_ (65% v v^−1^), and the volume was made up with type 1 water (1 mL) before ICP-MS analysis. Analyses were performed in triplicate, and the percentage of encapsulated Cu^2+^ ions in CS NPs was determined using Equation (1):Encapsulation efficiency (%) = 100 × (Total Cu^2+^ − Free Cu^2+^)/Total Cu^2+^(1)

### 4.4. Biological Material and Treatments

The study was carried out using the leaf disc method [40]. Urediniospores of *H. vastatrix* were obtained from naturally infected coffee leaves in the field of the Rural Development Institute of Parana—IAPAR-EMATER (IDR -Parana, Londrina, Parana, BR). The leaves containing rust were stored in paper bags throughout the collection procedure (Appendix A) and were subsequently sent to the Laboratory of Plant Ecophysiology of the Department of Animal and Plant Biology at the State University of Londrina (UEL, Londrina, Parana, BR). Previously, 1 mL Pasteur pipettes were adapted by removing the graduated part to keep only the base of the pipette, which served as a collection container for urediniospores. With the aid of adapted pipettes, the urediniospores of *H. vastatrix* were detached from the abaxial surface of the leaves by scraping the leaf blade and then storing it in 15 mL falcon tubes at −80 °C until use.

Leaf discs were obtained from *Coffea arabica* cv. IPR 100 seedlings: a cultivar susceptible to coffee rust [41]. Seedlings with six pairs of leaves were transplanted from tubes to 20 × 26 cm (2 kg) nursery plant polybags filled with previously sterilized clay-textured soil. During the development in the nursery (50% shading and without ambient temperature control) at the IDR -Paraná, the seedlings were treated with pesticides recommended for coffee crops (when necessary) and fertilized via a substrate with Osmocote Plus^®^ (15% N; 9% P_2_O_5_; 12% K_2_O; 1.3% Mg; 6% S; 0.05% Cu; 0.46% Fe; 0.06% Mn and 0.02% Mo). Six months before the beginning of the experiments, the seedlings were transferred to the greenhouse belonging to the Department of Animal and Plant Biology of the State University of Londrina (without shading and ambient temperature control). The fertilization was kept to maintain the seedlings, and sanitary control was carried out by manually removing plants and/or affected parts.

Different experiments were carried out under laboratory conditions. For each one, healthy and fully expanded leaves were randomly collected from the youngest branches of 60 coffee seedlings. The leaf discs were removed using an EVA perforator with a 16 mm (5/8”) diameter. Subsequently, the leaf discs were treated with distilled water as a control treatment (CONT), the suspension of unloaded CS NPs (NP), the suspension of CS NPs containing Cu^2+^ ions (NPCu), and a solution of Cu^2+^ ions (Cu). The stock formulations were diluted in water to obtain three different concentrations of CS and Cu (Table 1).

The procedure for treating and inoculating the leaf discs was the same for all the tests involving responses to infection with *H. vastatrix*, changing only the type of experimental unit and the number of treatments. With the aid of metallic tweezers, leaf discs were submerged individually for 5 s in a glass beaker containing 30 mL of the respective formulation. After the treatments, the leaf discs were transferred to transparent crystal polystyrene plastic boxes (Gerbox^®^; 11 cm width × 11 cm length × 3 cm height) containing a foam (2 cm thick) that was previously saturated with distilled water. Inoculation was performed 24 h after the treatment of the leaf discs. The required amount of urediniospores were separated into 0.6 mL microtubes and then incubated in a water bath at 40 °C for 10 min. After incubation, the microtubes were kept at room temperature for 2 h. A suspension of viable urediniospores was prepared at 1 mg mL^−1^ in distilled water for the inoculum. Under constant stirring, 0.025 mL of the inoculum was collected using an automatic micropipette and then added to the center of the abaxial face of each leaf disc (Appendix A). The plastic boxes with the inoculated leaf discs were kept in the dark for 48 h at 23 °C ± 1 °C. After the darkroom period, the lids of the boxes were opened for a while to reduce the droplet size before transferring to high luminosity. Finally, the boxes were closed and placed under the photosynthetically active radiation of 102 µmol m^−2^ s^−1^ and a 12 h photoperiod.

### 4.5. Experiment to Assess Phytotoxicity

The experiment consisted of the groups and concentrations described in Table 1, totaling 10 treatments. The leaf discs were not inoculated after treatment and were kept under natural light and temperature control (23 °C ± 1 °C). The visual analysis and the measurement of the pigment content of the leaf discs were carried out 40 days after the treatments to evaluate their potential phytotoxicity. Four replicates per treatment were used. Five leaf discs per repetition were collected, placed in a 15 mL falcon tube, and incubated with 5 mL of 80% (v v^−1^) acetone solution in 2.5 mmol L^−1^ sodium phosphate buffer (pH 7.8) for 15 days at 4 °C. Subsequently, the extracts were centrifuged at 1800× *g* for 5 min. The supernatant was collected, and the absorbance was read at 663.2, 646.8, and 470 nm. The concentrations of chlorophyll *a*, chlorophyll *b*, and carotenoids were calculated using the formulas proposed by Lichtenthaler and Buschmann [42]. The total chlorophyll content and total chlorophyll/carotenoids ratio were calculated.

### 4.6. Experiment to Evaluate Infection with Hemileia Vastatrix

The experiment consisted of the groups and concentrations described in Table 1, totaling 10 treatments. Leaf discs were inoculated following the methodology described above. To analyze the infection of leaf discs with *H. vastatrix*, the experimental units consisted of plastic boxes with 15 leaf discs, using 4 boxes per treatment. Disease assessment was performed using photographic records at 10, 15, 20, 25, 30, 35, and 40 days after infection (DAI). The photos were used to monitor the progression of the disease and to support subsequent assessments. The following parameters were evaluated in the experiment:

Incidence (at 15 DAI)—defined as the percentage of leaf discs in each replicate with disease symptoms;

Sporulation (at 25 DAI)—defined as the percentage of leaf discs in each replicate containing lesions with sporulation;

Severity (at 40 DAI)—defined as the percentage of injured leaf area at the end of the experiment. To obtain this variable, photos were analyzed using the ImageJ software [43] in order to determine the injured area of each leaf disc. The value of the variable in each repetition is the average of the leaf discs.

The pigment contents of the leaf discs were determined at 40 DAI, following the previously described methodology. The surface of each leaf disc was cleaned with distilled water and then dried with paper before incubation in an acetone buffer.

### 4.7. Experiment to Evaluate Oxidative Damage and Activity of Antioxidant Enzymes

Based on the results of experiments 4.5 and 4.6, the highest concentrations of NP (1 g L^−1^), NPCu, and Cu (5 mmol L^−1^) were removed from subsequent experiments to assess oxidative damage and enzymatic antioxidant activity, totaling 7 treatments.

The evaluations were made at 5 and 15 DAI. For each biochemical analysis, 4 replicates per treatment were used, with each replicate consisting of 3 leaf discs randomly selected from an individual plastic box. The leaf discs were weighed, immediately immersed in liquid nitrogen, and then stored at −80 °C until the analysis.

The content of H_2_O_2_ and MDA was determined as markers of oxidative damage, following the methodologies described by Alexieva et al. [44] and Camejo et al. [45], respectively.

To quantify the activity of antioxidant enzymes, the samples were homogenized in 1.5 mL of extraction buffer, consisting of 1 mmol L^−1^ EDTA in 0.1 mol L^−1^ potassium phosphate buffer (pH 7.5), supplemented with polyvinyl-poly-pyrrolidone (PVPP) 2% (w v^−1^). The extracts were centrifuged at 15,645× *g* (4 °C for 20 min). The activity of the enzyme ascorbate peroxidase (APX, EC 1.11.1.11) was determined according to the method proposed by Nakano and Asada [46], monitoring ascorbate consumption at 290 nm in the presence of H_2_O_2_. The catalase activity (CAT, EC 1.11.1.6) was determined according to Aebi et al. [47], Anderson et al. [48], and Peixoto et al. [49], following the reduction in absorbance of H_2_O_2_ at 240 nm. The peroxidase activity (POD, EC 1.11.1.7) was determined by the methodology proposed by Peixoto et al. [49], who monitored the increase in absorbance at 420 nm resulting from the oxidation of pyrogallol in the presence of H_2_O_2_. The superoxide dismutase activity (SOD, EC 1.15.1.1) was determined according to Giannopolitis and Ries [50], who measured the ability of the enzymatic extract to inhibit the photoreduction of nitroblue-tetrazolium chloride (NBT).

### 4.8. Experiment to Evaluate the Fluorescence of Chlorophyll a and the Activity of PAL and PPO Enzymes

The experiment was set up similarly to experiment 4.7, totaling the same 7 treatments. The same procedure was carried out for collecting and storing leaf discs until the analysis time. The evaluations were made with 5 and 15 DAI. Four replicates per treatment were used; each replication consisted of 3 leaf discs randomly selected from an individual plastic box.

Chlorophyll *a* fluorescence variables were measured in leaf discs using an OS1p portable fluorometer (Opti-Sciences, Hudson, NY, USA). The leaf discs were dark-adapted for 15 min using FL-DC clips, and the basal fluorescence (F_0_) was measured using a weak modulated light for 0.1 s (10% intensity). Then, the leaves were exposed to a light-saturating pulse (8250 µmol m^−2^ s^−1^) for 0.8 s to measure the maximum quantum yield of the photosystem II (F_v_/F_m_) [51]. The relative electron transport rate of photosystem II (rETR) was determined in light-adapted leaf discs exposed to the photosynthetically active radiation (PAR) of 102 μmol m^−2^ s^−1^. The basal fluorescence (F′) and the maximum fluorescence (F_m_′) of light-adapted leaf discs were determined before and after the exposure to the light-saturating pulse, respectively, and ΔF was calculated as the difference between F_m_′ and F′. The rETR was calculated according to the equation indicated by Baker [51]:rETR = ΔF/F_m_′ × PAR × 0.5 × 0.84 (2)

To quantify the activity of the phenylalanine ammonia-lyase (PAL, EC 4.3.1.5) and polyphenol oxidase (PPO, EC 1.10.3.1) enzymes, the samples were homogenized in 1.5 mL of the extraction buffer, consisting of 1 mmol L^−1^ EDTA in buffer 0.1 mol L^−1^ potassium phosphate (pH 7.5), supplemented with 2% polyvinyl-poly-pyrrolidone (PVPP) (w v^−1^) and 10 mmol L^−1^ dithiothreitol (DTT). The extract was then centrifuged at 15,645× *g* (4 °C for 20 min). PAL activity was determined by monitoring the increase in absorbance at 290 nm resulting from the non-oxidative deamination of L-phenylalanine to form trans-Cinnamic acid and ammonium. PPO activity was quantified by tracking the increase in absorbance at 420 nm resulting from the oxygen-dependent oxidation of monophenols. Both activities were determined according to the methodology proposed by Peixoto et al. [49].

### 4.9. Statistical Analyzes

For all experiments, the experimental design was completely randomized. First, the tests and graphical analyses of the residuals were performed to verify normality, homogeneity of variance, and independence. In the experiment to evaluate phytotoxicity, the results were submitted to regression analysis (*p* < 0.05). In the experiment to assess the infection with *H. vastatrix*, due to the impossibility of analyzing them using parametric statistics, the variables incidence, sporulation, and severity were analyzed using the Kruskal–Wallis test followed by the Dunn test (*p* < 0.05) for multiple comparisons of the mean ranks. The concentration of pigments was submitted to regression analysis (*p* < 0.05). In the other experiments to evaluate oxidative damage, enzymatic activity, and chlorophyll *a* fluorescence, the results were submitted to the analysis of variance (ANOVA) by the F test, and, when significant, the means were compared by the Scott-Knott test (*p* < 0.05). All analyzes were performed using the R statistical program [52], using the packages easyanova, ExpDes.pt, asbio, PMCMR, and agricolae.

## 5. Conclusions

The phytotoxicity of treatments is directly related to the applied dose, with the highest concentrations of all treatments being harmful to plant metabolism. While the treatments containing CS NPs (especially NPCu) stimulated the synthesis of chlorophyll *a* while significantly increasing the total chlorophyll content in the leaf discs, the treatment Cu reduced the carotenoid content in all the concentrations applied. The treatments NP (1 g L^−1^), NPCu (1.25, 2.5, and 5 mmol L^−1^), and Cu (1.25, 2.5, and 5 mmol L^−1^) effectively reduced the incidence, sporulation, and severity of *H. vastatrix*-induced disease. However, the NPCu treatments stood out as maintaining the highest levels of pigments compared to Cu, NP, and the control. Although both concentrations (1.25 and 2.5 mmol L^−1^) of NPCu and Cu treatments increased antioxidant responses in leaf discs, only NPCu treatments could mitigate oxidative damage over the days of infection. In addition, treatment with Cu (2.5 mmol L^−1^) further compromised the activity of photosystem II of the leaf discs, evidencing the phytotoxic effect caused by high Cu^2+^ ion concentrations. At low concentrations (1.25 mmol L^−1^), NPCu also showed higher bioactivity than Cu, reinforcing the highest efficiency of the nano-formulation.

## Figures and Tables

**Figure 1 antibiotics-12-00249-f001:**
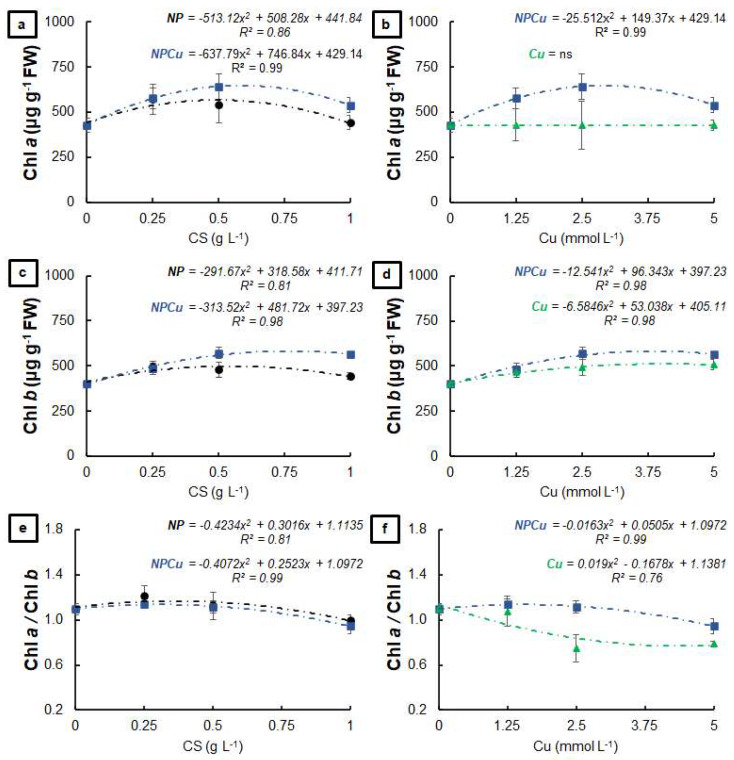
Chlorophyll *a* (**a**,**b**), chlorophyll *b* (**c**,**d**), and chlorophyll *a*/chlorophyll *b* ratio (**e**,**f**) of leaf discs of *Coffea arabica* cv. IPR 100 at 40 days after treatment with distilled water, as a control (CONT), chitosan nanoparticles without Cu^2+^ ions (NP), chitosan nanoparticles containing Cu^2+^ ions (NPCu), and free Cu^2+^ ions (Cu). The formulations were diluted in distilled water, obtaining the corresponding concentrations of chitosan (CS) in NP and NPCu treatments (0.25; 0.5; 1 g L^−1^) and of Cu^2+^ ions (1.25; 2.5; 5 mmol L^−1^) in NPCu and Cu treatments. Results are expressed as mean (*n* = 4) ± standard error. The model and coefficient of determination (R^2^) are also shown (*p* < 0.05). ns = not significant.

**Figure 2 antibiotics-12-00249-f002:**
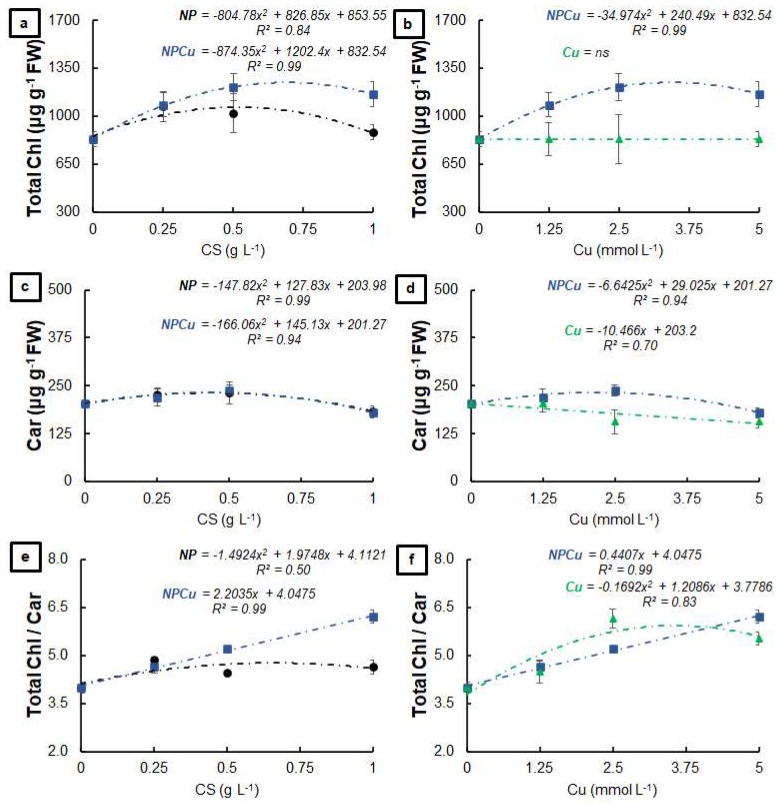
Total chlorophyll (**a**,**b**), carotenoids (**c**,**d**), and total chlorophyll/carotenoids ratio (**e**,**f**) of leaf discs of *Coffea arabica* cv. IPR 100 at 40 days after treatment with distilled water, as a control (CONT), chitosan nanoparticles without Cu^2+^ ions (NP), chitosan nanoparticles containing Cu^2+^ ions (NPCu), and free Cu^2+^ ions (Cu). The formulations were diluted in distilled water, obtaining the corresponding concentrations of chitosan (CS) in NP and NPCu treatments (0.25; 0.5; 1 g L^−1^) and of Cu^2+^ ions (1.25; 2.5; 5 mmol L^−1^) in NPCu and Cu treatments. Results are expressed as mean (*n* = 4) ± standard error. The model and coefficient of determination (R^2^) are also shown (*p* < 0.05). ns = not significant.

**Figure 3 antibiotics-12-00249-f003:**
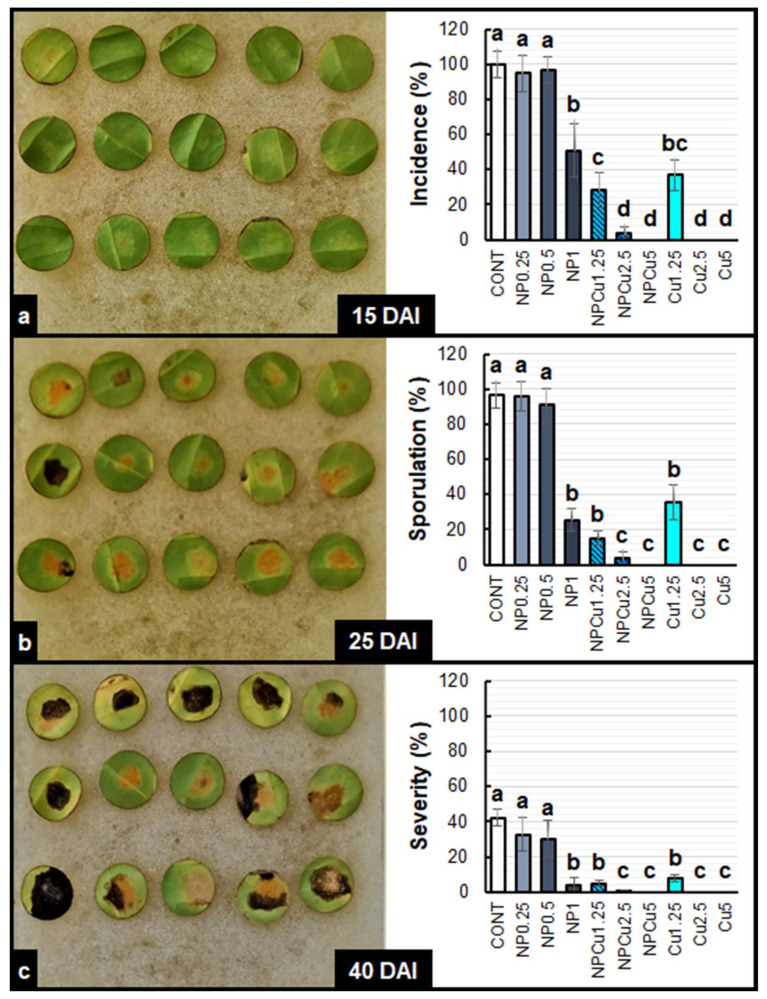
Incidence (**a**), Sporulation (**b**), and severity (**c**) of *Hemileia vastatrix* disease in leaf discs of *Coffea arabica* cv. IPR 100 at 15, 25, and 40 days after infection (DAI). Before infection, the discs were treated with distilled water as a control (CONT), chitosan nanoparticles without Cu^2+^ ions (NP), chitosan nanoparticles containing Cu^2+^ ions (NPCu), and free Cu^2+^ ions (Cu). The formulations were diluted in distilled water, obtaining the corresponding concentrations of chitosan (CS) in NP and NPCu treatments (0.25; 0.5; 1 g L^−1^) and of Cu^2+^ ions (1.25; 2.5; 5 mmol L^−1^) in NPCu and Cu treatments. Results are expressed as mean (*n* = 4) ± standard error. Equal letters in the columns indicate that there was no significant difference by Dunn test (*p* < 0.05).

**Figure 4 antibiotics-12-00249-f004:**
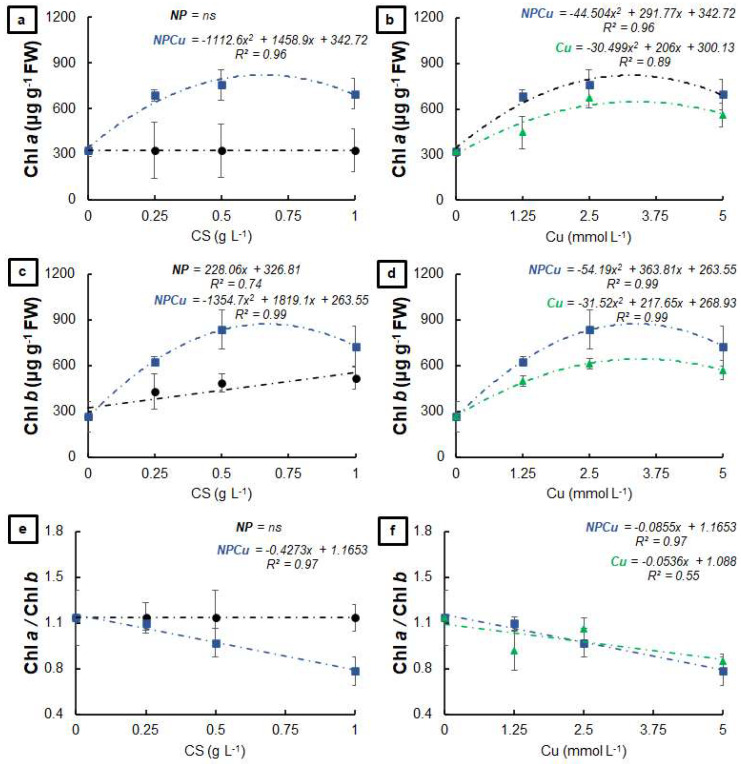
Chlorophyll *a* (**a**,**b**), chlorophyll *b* (**c**,**d**), and chlorophyll *a*/chlorophyll *b* ratio (**e**,**f**) of leaf discs of *Coffea arabica* cv. IPR 100 at 40 days after infection (DAI) with *H. vastatrix*, Before infection, the discs were treated with distilled water as a control (CONT), chitosan nanoparticles without Cu^2+^ ions (NP), chitosan nanoparticles containing Cu^2+^ ions (NPCu), and free Cu^2+^ ions (Cu). The formulations were diluted in distilled water, obtaining the corresponding concentrations of chitosan (CS) in NP and NPCu treatments (0.25; 0.5; 1 g L^−1^) and of Cu^2+^ ions (1.25; 2.5; 5 mmol L^−1^) in NPCu and Cu treatments. Results are expressed as mean (*n* = 4) ± standard error. The model and coefficient of determination (R^2^) are also shown (*p* < 0.05). ns = not significant.

**Figure 5 antibiotics-12-00249-f005:**
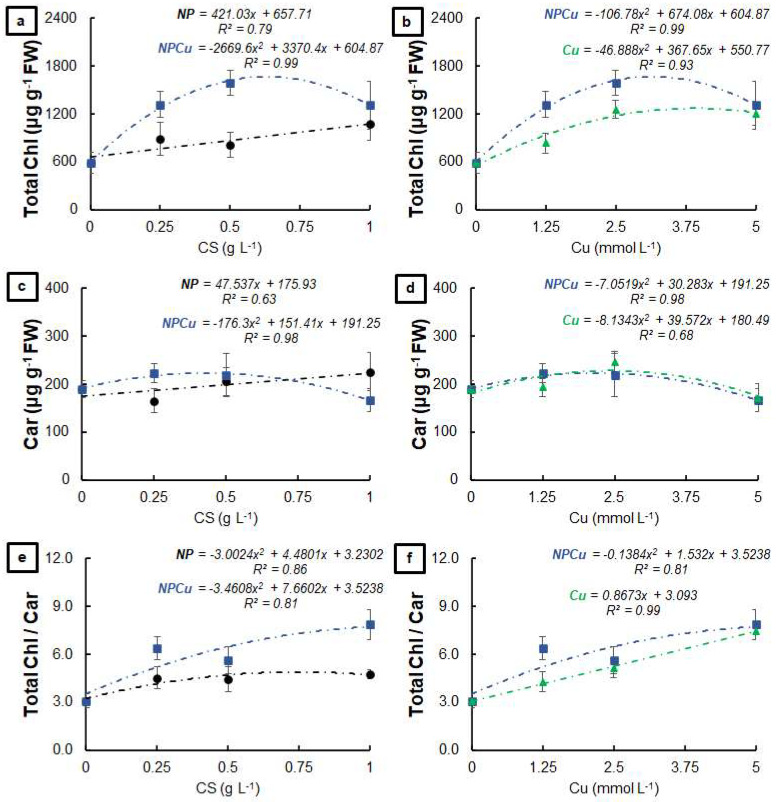
Total chlorophyll (**a**,**b**), carotenoids (**c**,**d**), and total chlorophyll/carotenoids ratio (**e**,**f**) of leaf discs of *Coffea arabica* cv. IPR 100 at days after infection (DAI) with *H. vastatrix*. Before infection, the discs were treated with distilled water as a control (CONT), chitosan nanoparticles without Cu^2+^ ions (NP), chitosan nanoparticles containing Cu^2+^ ions (NPCu), and free Cu^2+^ ions (Cu). The formulations were diluted in distilled water, obtaining the corresponding concentrations of chitosan (CS) in NP and NPCu treatments (0.25; 0.5; 1 g L^−1^) and of Cu^2+^ ions (1.25; 2.5; 5 mmol L^−1^) in NPCu and Cu treatments. Results are expressed as mean (*n* = 4) ± standard error. The model and coefficient of determination (R^2^) are also shown (*p* < 0.05).

**Figure 6 antibiotics-12-00249-f006:**
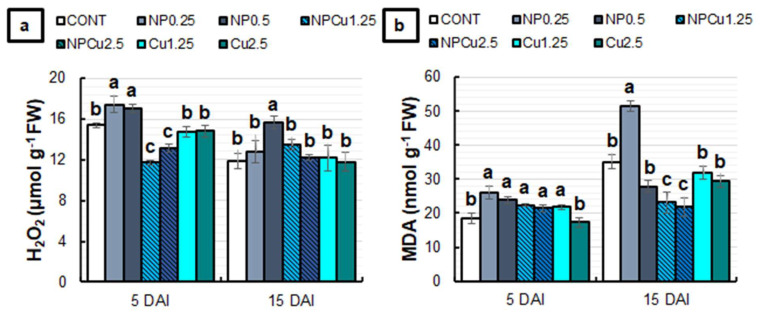
Hydrogen peroxide (H_2_O_2_) (**a**) and malondialdehyde (MDA) (**b**) content of leaf discs of *Coffea arabica* cv. IPR 100 at 5 and 15 days after infection (DAI) with *H. vastatrix*. Before infection, the discs were treated with distilled water as a control (CONT), chitosan nanoparticles without Cu^2+^ ions (NP), chitosan nanoparticles containing Cu^2+^ ions (NPCu), and free Cu^2+^ ions (Cu). The formulations were diluted in distilled water, obtaining the corresponding concentrations of chitosan (CS) in NP and NPCu treatments (0.25; 0.5 g L^−1^) and of Cu^2+^ ions (1.25; 2.5 mmol L^−1^) in NPCu and Cu treatments. Results are expressed as mean (*n* = 4) ± standard error. Equal letters in the columns indicate that there was no significant difference in the Scott–Knott test (*p* < 0.05) at the same time point.

**Figure 7 antibiotics-12-00249-f007:**
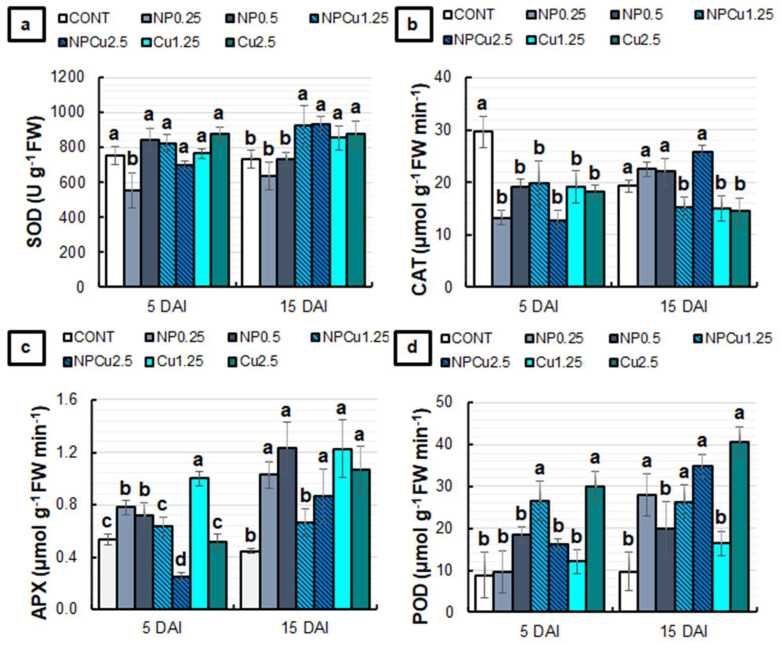
The activity of Superoxide dismutase (SOD) (**a**), Catalase (CAT) (**b**), Ascorbate peroxidase (APX) (**c**), and Peroxidase (POD) (**d**) of leaf discs of *Coffea arabica* cv. IPR 100 at 5 and 15 days after infection (DAI) with *H. vastatrix*. Before infection, the discs were treated with distilled water as a control (CONT), chitosan nanoparticles without Cu^2+^ ions (NP), chitosan nanoparticles containing Cu^2+^ ions (NPCu), and free Cu^2+^ ions (Cu). The formulations were diluted in distilled water, obtaining the corresponding concentrations of chitosan (CS) in NP and NPCu treatments (0.25; 0.5 g L^−1^) and of Cu^2+^ ions (1.25; 2.5 mmol L^−1^) in NPCu and Cu treatments. Results are expressed as mean (*n* = 4) ± standard error. Equal letters in the columns indicate that there was no significant difference in the Scott–Knott test (*p* < 0.05) at the same time point.

**Figure 8 antibiotics-12-00249-f008:**
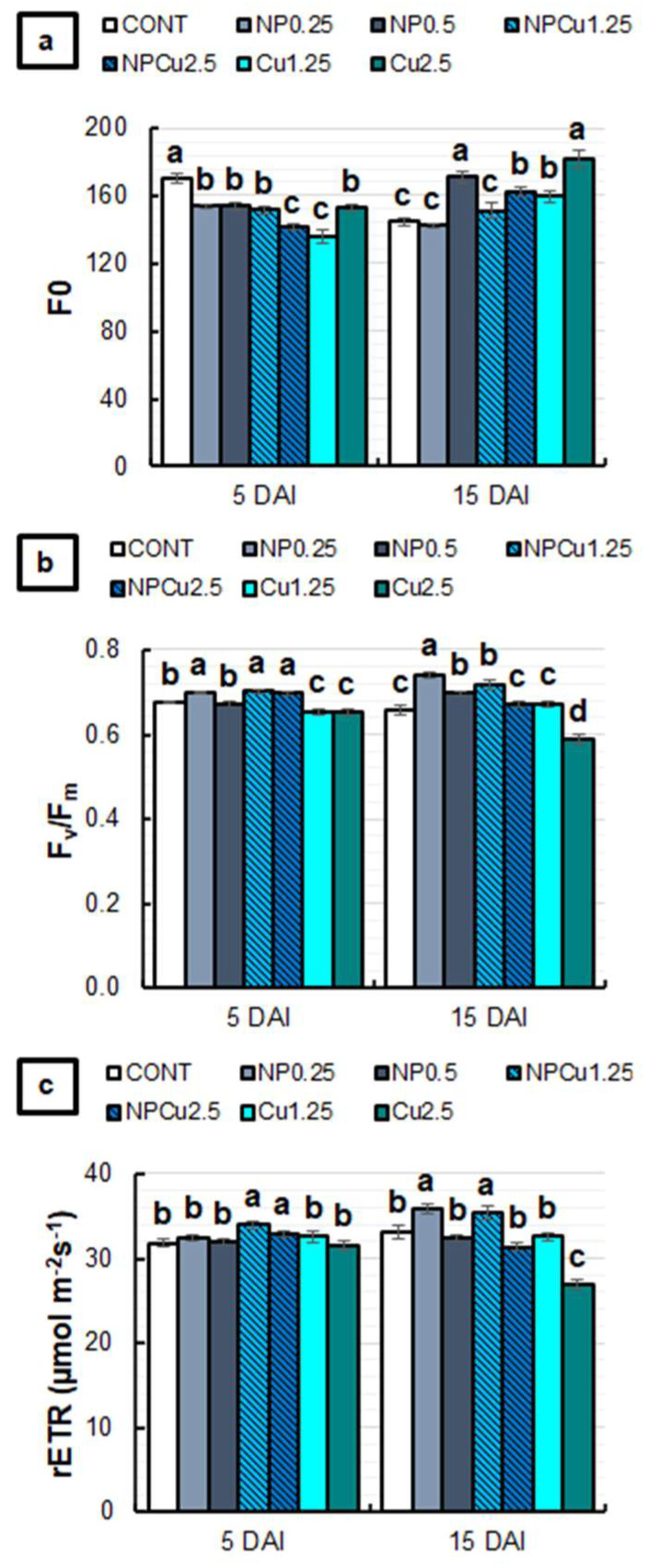
F_0_ (**a**), F_v_/F_m_ (**b**), and rETR (**c**) of leaf discs of *Coffea arabica* cv. IPR 100 at 5 and 15 days after infection (DAI) with *H. vastatrix*. Before infection, the discs were treated with distilled water as a control (CONT), chitosan nanoparticles without Cu^2+^ ions (NP), chitosan nanoparticles containing Cu^2+^ ions (NPCu), and free Cu^2+^ ions (Cu). The formulations were diluted in distilled water, obtaining the corresponding concentrations of chitosan (CS) in NP and NPCu treatments (0.25; 0.5 g L^−1^) and of Cu^2+^ ions (1.25; 2.5 mmol L^−1^) in NPCu and Cu treatments. Results are expressed as mean (*n* = 4) ± standard error. Equal letters in the columns indicate that there was no significant difference in the Scott–Knott test (*p* < 0.05) at the same time point.

**Figure 9 antibiotics-12-00249-f009:**
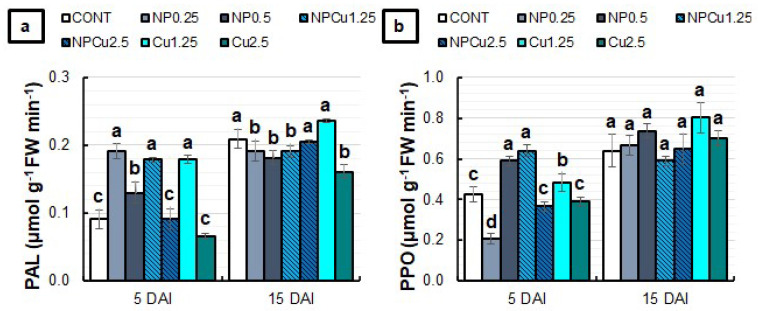
The activity of phenylalanine ammonia-lyase (PAL) (**a**) and polyphenol oxidase (PPO) (**b**) enzymes of leaf discs of *Coffea arabica* cv. IPR 100 at 5 and 15 days after infection (DAI) with *H. vastatrix*. Before infection, the discs were treated with distilled water as a control (CONT), chitosan nanoparticles without Cu^2+^ ions (NP), chitosan nanoparticles containing Cu^2+^ ions (NPCu), and free Cu^2+^ ions (Cu). The formulations were diluted in distilled water, obtaining the corresponding concentrations of chitosan (CS) in NP and NPCu treatments (0.25; 0.5 g L^−1^) and of Cu^2+^ ions (1.25; 2.5 mmol L^−1^) in NPCu and Cu treatments. Results are expressed as mean (*n* = 4) ± standard error. Equal letters in the columns indicate that there was no significant difference in the Scott–Knott test (*p* < 0.05) at the same time point.

**Table 1 antibiotics-12-00249-t001:** Treatments applied to leaf discs and their corresponding chitosan (CS) and Cu^2+^ concentrations.

Experimental Group	CS (g L^−1^)	Cu^2+^ (mmol L^−1^)
CONT (Control)	0.00	0.00
NP0.25	0.25	0.00
NP0.5	0.50	0.00
NP1	1.00	0.00
NPCu1.25	0.25	1.25
NPCu2.5	0.50	2.50
NPCu5	1.00	5.00
Cu1.25	0.00	1.25
Cu2.5	0.00	2.50
Cu5	0.00	5.00

## Data Availability

The data supporting the reported results can be found through a direct request to the corresponding authors.

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
