# Peer review of "Nanoencapsulation Boosts the Copper-Induced Defense Responses of a Susceptible Coffea arabica Cultivar against Hemileia vastatrix"

_antibiotics, 2023, doi:10.3390/antibiotics12020249_

Round 1
Reviewer 1 Report
Nanoencapsulation boosts the copper-induced defense responses of a susceptible Coffea arabica cultivar against Hemileia vastatrix is well written and authors covered almost all the characterization parameters. Some changes have to be done.
line 57 write in the market instead of on the market
apply std dev in all the graphs or images
need some grammatical improvement in the manuscript.
modify conclusion section, add some more results
Author Response
Reviewer #1:
Nanoencapsulation boosts the copper-induced defense responses of a susceptible Coffea arabica cultivar against Hemileia vastatrix is well written and authors covered almost all the characterization parameters. Some changes have to be done.
Response: Thank you very much for your comments and suggestions regarding the manuscript. We believe that greatly enhanced the quality of the manuscript.
#1 line 57 write in the market instead of on the market
Response: Corrected.
#2 apply std dev in all the graphs or images
Response: We included this information in the manuscript.
#3 need some grammatical improvement in the manuscript.
Response: We agree. We provide a textual revision to improve the grammar of the manuscript.
#4 modify conclusion section, add some more results
Response: We have improved the conclusion section including some more results.
Reviewer 2 Report
The authors of the article, "Nanoencapsulation boosts the copper-induced defense re- 2 sponses of a susceptible Coffea arabica cultivar against Hemileia 3 vastatrix", presented all the findings in an excellent way. From my point of view, the conclusion may be enhanced with more points. Language correction should be checked.
Author Response
Reviewer#2:
The authors of the article, "Nanoencapsulation boosts the copper-induced defense responses of a susceptible Coffea arabica cultivar against Hemileia vastatrix", presented all the findings in an excellent way. From my point of view, the conclusion may be enhanced with more points. Language correction should be checked.
Response: Thank you very much for your comments and suggestions regarding the manuscript. We believe that greatly enhanced the quality of the manuscript. We have improved the conclusion section including some more results. Also, we provide a textual revision to improve the grammar of the manuscript.
Reviewer 3 Report
The authors reported on the defense response induced by the Cu2+ ion-containing chitosan nanoparticles on leaf discs of Coffea arabica cv. IPR 100 infected with H. vastatrix.
The reported work is relatively new and the study on the defense response is systematic. Therefore, I recommend the publication on Antibiotics upon the following conditions are well addressed.
[1] The resolution of the Figures 1,2,4&5 can be improved. Also, there are no clearly descriptions on the “Cu=ns” and “NP=ns” for the annotations.
[2] The Figures 1,2,4&5 are very similar and the Figures 6–9 are also very similar. At least the colors of the lines and bars can be modified for better contrast.
[3] For the evaluation of chlorophyll a fluorescence, the authors only reported on the change of the fluorescence with different testing conditions. However, there is deficiency of the scientific explanations for why the change of the fluorescence value and what are the health status of the leaf discs implied by the results of the F0, Fv/Fm and rETR.
Author Response
Reviewer #3:
The authors reported on the defense response induced by the Cu2+ ion-containing chitosan nanoparticles on leaf discs of Coffea arabica cv. IPR 100 infected with H. vastatrix. The reported work is relatively new and the study on the defense response is systematic. Therefore, I recommend the publication on Antibiotics upon the following conditions are well addressed.
Response: Thank you very much for your comments and suggestions regarding the manuscript. We believe that greatly enhanced the quality of the manuscript.
#1 The resolution of the Figures 1,2,4&5 can be improved. Also, there are no clearly descriptions on the “Cu=ns” and “NP=ns” for the annotations.
Response: We agree. We have improved the resolution of Figures 1,2,4 e 5 and included the description of “Cu=ns” and “NP=ns” as not significant in the captions of the respective figures.
#2 The Figures 1,2,4&5 are very similar and the Figures 6–9 are also very similar. At least the colors of the lines and bars can be modified for better contrast.
Response: We changed the colors of the bars and lines to have a clearer view and better contrast among treatments.
#3 For the evaluation of chlorophyll a fluorescence, the authors only reported on the change of the fluorescence with different testing conditions. However, there is deficiency of the scientific explanations for why the change of the fluorescence value and what are the health status of the leaf discs implied by the results of the F0, Fv/Fm and rETR.
Response: The parameters F0, Fv/Fm, and rETR are used to estimate the occurrence of stress in plants through a fast and non-destructive way. In our specific case, the data of these parameters allowed us to observe the direct action of H. vastatrix and/or the effect of high concentrations of the formulations on the activity of photosystem II. The maximum quantum yield of photosystem II (Fv/Fm) is measured in dark-adapted leaves. The optimal values of Fv/Fm are near 0.8; thus, in this study, lowered values indicate the inhibitory effect in photosystem II. The concentrations of all formulations could control H. vastatrix and prevent the decrease in photosystem II activity. The greater the disease control and, consequently, the health of the leaf disc, the greater the Fv/Fm value. The relative electron transport rate of photosystem II (rETR) can be additionally used to indicate the H. vastatrix action and/or disease control as well as the inhibitory effects over the activity of photosystem II. However, a reduction in Fv/Fm and rETR was observed in the treatment containing Cu2+ ions (2.5 mmol L-1), indicating that in this treatment an additional inhibitory effect in photosystem II was caused both by the stress induced by the concentration applied (2.5 mmol L-1). Regarding F0, increased values of this parameter indicate damage to photosystem II. The decrease in the Fv/Fm ratio, concomitant with an increase in F0 (Cu 2.5 mmol L-1), indicates damage to photosystem II, which has been observed in plants under different types of stress. Also, we provided more information about the chlorophyll fluorescence analysis in the Material and Methods.
Reviewer 4 Report
The paper entitled" Nanoencapsulation boosts the copper-induced defense responses of a susceptible Coffea arabica cultivar against Hemileia vastatrix" is a very interesting well-written paper that discussed a very important plant disease and tries to solve its problem with minimum impact on health and the environment.
The result and discussion look comprehensive and provide all the needed details of the experiment.
Although the paper is large, it provides all the required data to be repeated and I accept it as it is.
Author Response
Reviewer #4:
The paper entitled" Nanoencapsulation boosts the copper-induced defense responses of a susceptible Coffea arabica cultivar against Hemileia vastatrix" is a very interesting well-written paper that discussed a very important plant disease and tries to solve its problem with minimum impact on health and the environment. The result and discussion look comprehensive and provide all the needed details of the experiment. Although the paper is large, it provides all the required data to be repeated and I accept it as it is.
Response: Thank you very much for your comments regarding the manuscript. Your review enhances the quality of the manuscript.